# Oxygen Vacancies Defective La_2_Ti_2_O_7_ Nanosheets Enhanced Photocatalytic Activity of Hydrogen Evolution under Visible Light Irradiation

**DOI:** 10.3390/molecules28155792

**Published:** 2023-07-31

**Authors:** Zhigang Wang, Hongliang Yu, Zhuoyuan Liu

**Affiliations:** School of Chemistry and Chemical Engineering, Hubei Polytechnic University, Huangshi 435003, China; 18247500736@163.com (H.Y.); 13886963917@163.com (Z.L.)

**Keywords:** oxygen vacancy, La2Ti2O7 nanosheets, photocatalysis, hydrogen evolution, visible light

## Abstract

A novel and efficient technique has been designed for the creation of oxygen vacancies on La_2_Ti_2_O_7_ (LTO) nanosheets. This is achieved via a controlled solid-state reaction between NaBH_4_ and LTO nanosheets. Transmission electron microscopy (TEM) analyses expose that these processed LTO specimens possess a unique crystalline core/amorphous shell structure, represented as La_2_Ti_2_O_7_@La_2_Ti_2_O_7-x_. According to X-ray photoelectron spectroscopy (XPS) observations, there is a notable correlation between the reaction time, temperature, and the concentration of oxygen vacancies. The concentration of these vacancies tends to increase along with the reaction time and temperature. Concurrently, UV-Visible spectra and photocatalytic tests reveal a significant impact of oxygen vacancies on the LTO surface on both light absorption and photocatalytic functionality. Most notably, the LTO nanosheets with engineered oxygen vacancies have demonstrated an exceptional photocatalytic capacity for hydrogen production under visible light. The maximal activity recorded was an impressive 149 μmol g^−1^ h^−1^, which is noticeably superior to the performance of the pristine La_2_Ti_2_O_7_.

## 1. Introduction

While solar hydrogen production with semiconductor photocatalysts is widely considered as a promising solution to global energy and environmental challenges, there is an extensive array of photocatalysts to choose from [1,2,3,4,5,6,7,8,9]. Among these, perovskite metal oxides such as La_2_Ti_2_O_7_ are particularly promising. The reason for their appeal lies in their unique electronic and optical properties, photochemical stability, low cost, and high catalytic efficiency [10,11,12,13,14,15]. These factors have drawn a significant amount of attention towards them, further aided by their layered structures, which facilitate the separation of H_2_ evolution sites.

The layered perovskite La_2_Ti_2_O_7_ nanosheets embody the best of both perovskite and layered structures, thereby exhibiting superior photocatalytic activity and high stability in tasks like organic decomposition, water splitting, and CO_2_ reduction [16,17,18,19,20,21,22,23,24,25,26,27,28,29]. However, La_2_Ti_2_O_7_ is not without its limitations. Its wide band gap of 3.5 eV restricts its activity to just under UV light irradiation. Also, its ionic character at room temperature impairs carrier transport and diminishes the efficiency of charge carrier separation. To circumvent these issues, numerous attempts have been made to broaden the photo-absorption of La_2_Ti_2_O_7_, using methods like doping and amalgamation with another semiconductor [30,31,32,33,34,35]. Despite these efforts, the widespread application of La_2_Ti_2_O_7_ remains significantly constrained due to the high recombination rate of photogenerated electron–hole pairs. Therefore, enhancing the photocatalytic efficiency of La_2_Ti_2_O_7_ remains a challenging task that the scientific community needs to address.

Oxygen vacancies, prevalent defects in oxide semiconductors, present potent strategies for regulating the charge separation process and manipulating the electronic structure of semiconductors [36,37,38,39,40,41]. Recently, it has been noted that these vacancies, also known as oxygen vacancy defects (ODs), can augment solar light harvesting in semiconductors. This enhancement is achieved by narrowing the band gap and acting as active sites to boost carrier separation efficiency, thereby resulting in improved water-splitting efficiency. Incorporating surface oxygen vacancies can generate new defect levels between the La_2_Ti_2_O_7_’s valence band maximum (VBM) and conduction band minimum (CBM). These new levels are advantageous as they draw more towards visible-light absorption, thereby augmenting solar harvesting. Moreover, it is proposed that a high concentration of defects amassing at the contact interface of composites is advantageous for forming ohmic contact. This is because the surface oxygen defects can serve as adsorption sites and trapping centers for photoinduced charges. Therefore, the management of oxygen defects on the surface of the photocatalyst could be instrumental in enhancing its photocatalytic properties.

In this current study, we have successfully synthesized novel oxygen vacancy-induced La_2_Ti_2_O_7_ nanosheets. This was achieved using a carefully controlled solid-state reaction between NaBH_4_ and crystalline La_2_Ti_2_O_7_. X-ray photoelectron spectroscopy (XPS) analysis provided substantial confirmation of the presence of surface oxygen vacancies. We evaluated the photocatalytic activity of these oxygen-deficient La_2_Ti_2_O_7_ nanosheets under both simulated sunlight and visible light for hydrogen production. These modified nanosheets showcased significant advantages when compared to their pure La_2_Ti_2_O_7_ counterparts.

## 2. Results and Discussion

Figure 1 presents the X-ray diffraction (XRD) patterns of the oxygen vacancy defective La_2_Ti_2_O_7_ nanosheet prepared at varying solid-state reaction temperatures. These patterns align well with the standard card of the Joint Committee on Powder Diffraction Standards (JCPDS) No. 70-0903. The characteristic diffraction peaks at 21.13°, 28.16°, 29.85°, 32.15°, 40.01°, 46.53°, and 58.01° correspond, respectively, to the (021), (201), (211), (041), (202), (−410), and (223) planes of La_2_Ti_2_O_7_. This correspondence underscores that pure La_2_Ti_2_O_7_ was successfully synthesized via the hydrothermal process. The intensity of the diffraction peaks further attests to the high crystallinity of these oxygen-deficient La_2_Ti_2_O_7_ (ODLTO) samples, comparable to the pristine La_2_Ti_2_O_7_ nanosheets. Nevertheless, an observable slight broadening of the main peak accompanies an increase in the reaction temperature [42]. This phenomenon could be attributed to the presence of oxygen vacancies, potentially leading to lattice strains and a reduction in the crystallite size.

Figure 2 depicts transmission electron microscopy (TEM) images that confirm the unchanged size and form of LTO nanosheets, both pre- and post-sodium borohydride (NaBH_4_) treatment. The average width of these LTO nanosheets measures approximately 200 nanometers (nm), as shown in Figure 2a,b. Examining the detailed structural aspects of the LTO nanosheets in high-resolution TEM (HRTEM) images provides more comprehensive insights. Prior to reduction treatment, the LTO nanosheets are characterized by a robust crystalline structure, with clearly distinguishable lattice features spanning the entirety of the particles, as shown in Figure 2c. Upon increasing the treatment temperature to 425 °C for a duration of 90 min, as displayed in Figure 2d, it becomes apparent that the nanocrystals’ core still retains its robust crystalline nature. Moreover, the lattice plane distances within this crystalline core are 0.28 nm, matching the [002] lattice plane of cubic LTO nanosheets [42]. The findings from HRTEM imagery substantiate that all these oxygen-deficient LTO (ODLTO) samples share a common characteristic: a core/shell structure comprised of LTO and oxygen-deficient LTO. 

To delve deeper into the surface compositions and the chemical state of La_2_Ti_2_O_7_ and its oxygen vacancy-defective variant, we conducted X-ray photoelectron spectroscopy (XPS) analyses, the results of which are presented in Figure 3. According to the survey XPS spectrum in Figure 3a, both LTO and its oxygen vacancy-defective version contain elements La, Ti, and O, with no traces of additional impurities. Figure 3b–d highlight the high-resolution spectra of La 3d, Ti 2p, and O 1s for both pristine LTO and the ODLTO-425 sample. The analysis reveals that in comparison to the spectra of the original LTO, the peaks for La and Ti progressively shift towards higher binding energy as the reaction temperature increases. Among the samples, the ODLTO-425 displays the most significant shift, exhibiting an approximate 0.3 eV change following the reduction treatment. Specifically, in Figure 3b, the La 3d spectra validate the existence of La^3+^ species in the ODLTO. The binding energy of La 3d_5/2_ for the untouched LTO was found at 833.5 eV. Meanwhile, for the ODLTO-425 sample, it rose slightly to 833.9 eV, mirroring the binding energy shift observed for Ti^4+^ species within the LTO, thus implying the formation of an oxygen vacancy in the oxygen-deficient version of LTO. The O 1s spectra for LTO samples can be distinguished into two peaks at 529.2 eV and 531.1 eV, respectively. The peak at 531.1 eV typically corresponds to the oxygen molecules adsorbed onto the LTO surface, while the 529.2 eV peak is linked to the Ti-O bonds integral to LTO’s crystalline structure (orange line in Figure 3d). In the case of the ODLTO-425 sample, a novel peak for O^2−^ is identified at 531.4 eV in the broad O 1s region, in addition to the peaks mentioned earlier [43]. This newly identified peak is indicative of an oxygen defect on the LTO surface.

The optical attributes and band structure details of the prepared samples were examined using UV-Vis absorption spectroscopy and X-ray photoelectron spectroscopy (XPS). The UV-Vis spectrum, as shown in Figure 4a, demonstrates that pristine LTO nanosheets can absorb ultraviolet radiation only up to wavelengths of 300 nm. However, in contrast, the ODLTO-425 sample exhibits an additional absorption band that extends beyond 400 nm, reaching into the wide-spectrum visible light zone [44]. This increased absorption can be attributed to the formation of reduced Ti sites, such as Ti^3+^, which occur because of increasing oxygen vacancies. The band structure was elucidated via a Tauc plot (Figure 4b) and valence band X-ray photoelectron spectroscopy (VB-XPS). The bandgap and VB positions for the pristine LTO and ODLTO-425 samples were found to be 3.18 and 2.94 eV and 1.87 and 2.09 V vs. NHE, respectively (Figure 4c). Based on these observations, the conduction band potentials (E_CB_) for pristine LTO and ODLTO-425 samples were determined to be −1.31 and −0.85 V vs. NHE (Figure 4c). Figure 4d illustrates the band structures of both pristine LTO and the ODLTO-425 sample. Therefore, the amplified visible light absorption in the ODLTO nanosheets can be linked to the electron transition, either from the valence band to the new defect level or from the defect level to the conduction band [45].

The photocatalytic efficiency of lithium titanate (LTO) samples was evaluated by measuring their hydrogen (H_2_) production from water using visible light irradiation. We utilized 20 mg of the samples, each loaded with 1.0 wt % platinum (Pt) and dispersed these in a 100 mL solution of 25% methanol in water. The source of visible light was a four LED lamp. As represented in Figure 5a, pristine LTO nanosheets exhibited a modest H_2_ production rate of merely around 3 μmol g^−1^ h^−1^. However, post-treatment with sodium borohydride (NaBH_4_) led to a noticeable boost in the H_2_ production rates of the resulting oxidized LTO (ODLTO) samples. Most notably, the gray-colored ODLTO-425 sample achieved a maximum H_2_ production rate soaring to an impressive 149 g^−1^ h^−1^ under visible light exposure. Further experimentation with sacrificial agents yielded the results displayed in Figure 5b. The stability of ODLTO-425 was also tested via recycling experiments. As shown in Figure 5c, the ODLTO-425 sample demonstrated little to no reduction in its photocatalytic performance in H_2_ generation, even after ten recycles, underscoring its commendable stability in water-splitting operations. To delve into the correlation between the light absorption of modified LTO samples and its photocatalytic activity, we carried out incident-photon-to-current conversion efficiency (IPCE) measurements on both the original LTO and the optimized ODLTO-425 photoanodes at −1.0 V (see Figure 5d). Compared to the pristine LTO, the ODLTO-425 sample showcased a significantly superior photocatalytic performance across both the ultraviolet (UV) and visible light spectrum. This implies that the heightened photocatalytic activity of the ODLTO-425 sample originates from an improved separation of photogenerated charge carriers in the UV-visible region.

To gain a deeper understanding of how oxygen vacancies influence the hydrogen evolution activity of LTO and the series of ODLTO-425 photocatalysts, we implemented photoluminescence (PL) and surface photovoltage (SPV) measurement techniques. PL spectra were first obtained to evaluate the recombination rate of photogenerated electron-hole pairs. Figure 6a presents the PL spectra of the freshly prepared photocatalysts. With an excitation wavelength set at 310 nm, all samples exhibit emission peaks roughly around 467 nm. However, when compared to LTO, both ODLTO-375 and ODLTO-425 samples show significantly diminished intensities of their emission peaks, attributable to the introduction of oxygen vacancies. This stark quenching of emission vividly illustrates that the induced oxygen vacancies have a potent facilitating effect on the separation of electron-hole pairs. To corroborate this inference, we employed surface photovoltage spectroscopy (SPV), an effective tool for revealing the separation and transport dynamics of photo-induced charge carriers, for the evaluation of these samples. As demonstrated in Figure 6b, the ODLTO-425 sample exhibits a more pronounced SPV response intensity than that of the pristine LTO. This indicates that oxygen vacancies favor the separation of photogenerated electron-hole pairs within LTO nanosheets, an observation that aligns seamlessly with the results derived from room-temperature photoluminescence (PL) spectroscopy (see Figure 6a). In conclusion, the findings emphatically suggest that the presence of oxygen vacancies in LTO nanosheets can significantly boost the separation efficiency of photo-excited electron–hole pairs, thereby improving their photocatalytic performance.

## 3. Experimental

### 3.1. Synthesis of Catalysts

La_2_Ti_2_O_7_ nanosheets were (LTO) synthesized via hydrothermal process. In a typical synthesis process, 2 mmol of La(NO_3_)_3_∙6H_2_O and 2 mmol of Ti(SO_4_)_2_ were dissolved in 10 mL ultrapure water with vigorous stirring. A total of 10 mL NaOH solution (2 M) was dropped into the above reaction solution. After magnetically stirring for 2 h, the obtained white mixture was transferred into a 50 mL Teflon-lined stainless-steel autoclave. The autoclave was sealed and heated at 240 °C for 24 h and then cooled to room temperature. The obtained products were collected via centrifugation, washed with distilled water and ethanol, respectively, and then dried at 70 °C for 6 h. Oxygen vacancy defective La_2_Ti_2_O_7_ nanosheets (ODLTO) were prepared via a solid-state reaction strategy: 100 mg of as-prepared La_2_Ti_2_O_7_ powder and 100 mg of NaBH_4_ were mixed in an agate mortar. Then, the mixture was put in a porcelain boat, placed in a tubular furnace, heated from room temperature to 375, 400, 425, and 450 °C under Ar at a heating rate of 5 °C min^−1^, and then held at the designed temperature for about 120 min. The as-prepared samples were denoted as ODLTO-T, where T is the reaction temperature (T = 375, 400, 425, and 450), respectively.

### 3.2. Characterization

The structure and crystallinity were analyzed via powder X-ray diffraction (XRD) on a D/MAX-RB diffractometer with Cu Kα radiation under the operation conditions of 40 kV and 50 mA. The absorption edges of samples were tested via a UV-vis spectrophotometer (DRS) with a UV-2550. The microstructure of the sample was examined via transmission electron microscopy (TEM) (JEM-2100F, Tokyo, Japan). X-ray photoelectron spectra (XPS) were recorded on a Thermo VG Multi-lab 2000 spectrometer with a monochromatic Al Kα source. Photoluminescence (PL) spectra characteristics were tested via fluorescence spectrophotometer (RF-5301 PC, Shimadzu, Japan) with an excitation wavelength of 320 nm. The lock-in amplifier-based surface photovoltage (SPV) consisted of a monochromatic light source, a lock-in amplifier (SR830-DSP) with a light chopper (SR540), 500 W Xe lamp (CHF-XM) acting as the light source.

### 3.3. Photocatalytic H_2_ Evolution Measurements

The H_2_ generation experiment was tested in a 435 mL sealed quartz three-necked Pyrex flask. The bottle was kept in a temperature-controlled air bath at 25 ± 0.5 °C with wind flow and was irradiated using a four 3 W LED lamp (420 nm). Then, appropriate amount of H_2_PtCl_6_ was added to make sure the photocatalysts were loaded with 1 wt % Pt using photodeposition method. The evolution gas was observed only under irradiation and analyzed using an on-line gas chromatograph (gas chromatograph (Shimadzu GC-2014C, Kyoto, Japan) loaded with a thermal conductive detector and ultrapure nitrogen as a carrier gas.

### 3.4. Photocatalytic H_2_ Evolution Measurements

A conventional three-electrode process was used to investigate the photoelectrochemical properties of samples in a quartz cell. The FTO photoanode deposited STO samples, Hg/Hg_2_Cl_2_, and Pt foil electrode acted as the working electrode, reference electrode, and counter electrode, respectively. A total of 1.0 M NaOH aqueous solution was used as the electrolyte. The photoanode was illuminated using a 300 W Xe lamp. The illuminated area was about 1 × 1 cm^2^.

## 4. Summary

In brief, this study presents a facile and comprehensive approach for generating oxygen vacancies on La_2_Ti_2_O_7_ nanosheets via a controllable solid-state reaction of NaBH_4_. The resulting LTO samples exhibit varying degrees of oxygen vacancy concentrations on their surfaces, which, in turn, affect their photocatalytic activities. Notably, the LTO samples with higher oxygen defect densities demonstrate significantly improved light absorption in the visible and infrared regions. The enhancement of photocatalytic performance for hydrogen evolution under visible light irradiation can be maximized through the improvement of charge separation of photogenerated charge carriers by achieving an optimal concentration of oxygen vacancies.

## Figures and Tables

**Figure 1 molecules-28-05792-f001:**
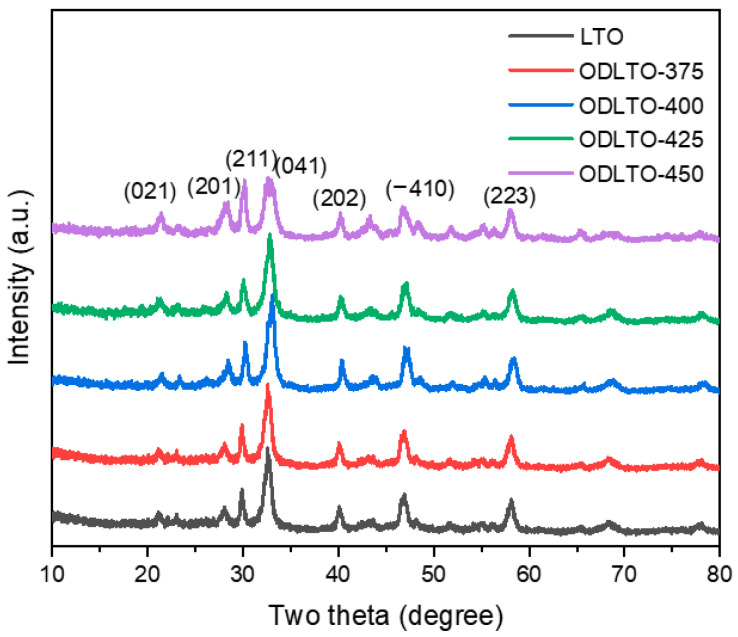
XRD patterns of the oxygen vacancy defective La_2_Ti_2_O_7_ nanosheet.

**Figure 2 molecules-28-05792-f002:**
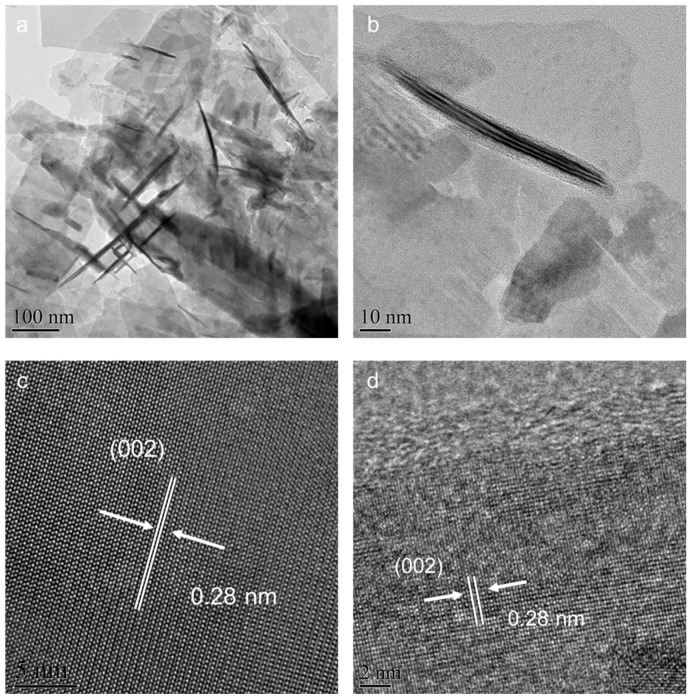
TEM images of (**a**) pristine LTO and (**b**) ODLTO-425 nanosheets; HRTEM images of (**c**) pristine LTO and (**d**) ODLTO-425 nanosheets.

**Figure 3 molecules-28-05792-f003:**
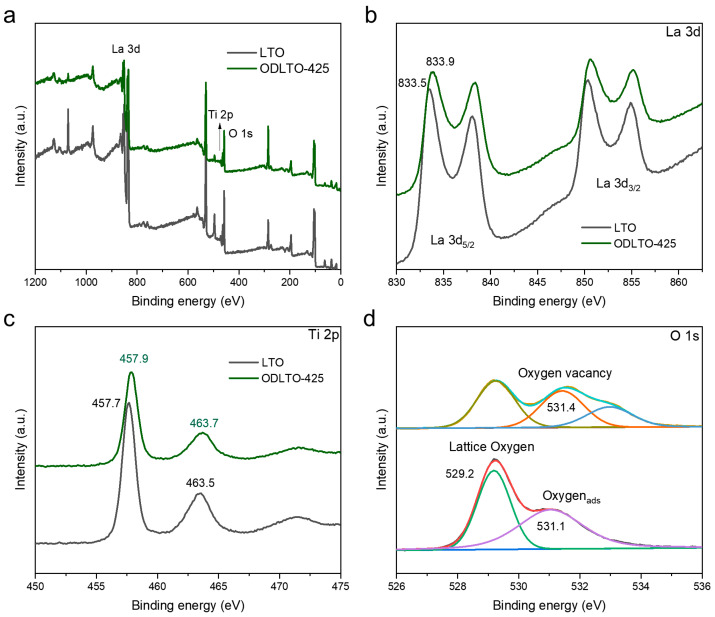
(**a**) XPS Survey spectra, (**b**) La 3d spectra, (**c**) Ti 2p spectra, and (**d**) O 1s spectra of pristine LTO and ODLTO-425 samples.

**Figure 4 molecules-28-05792-f004:**
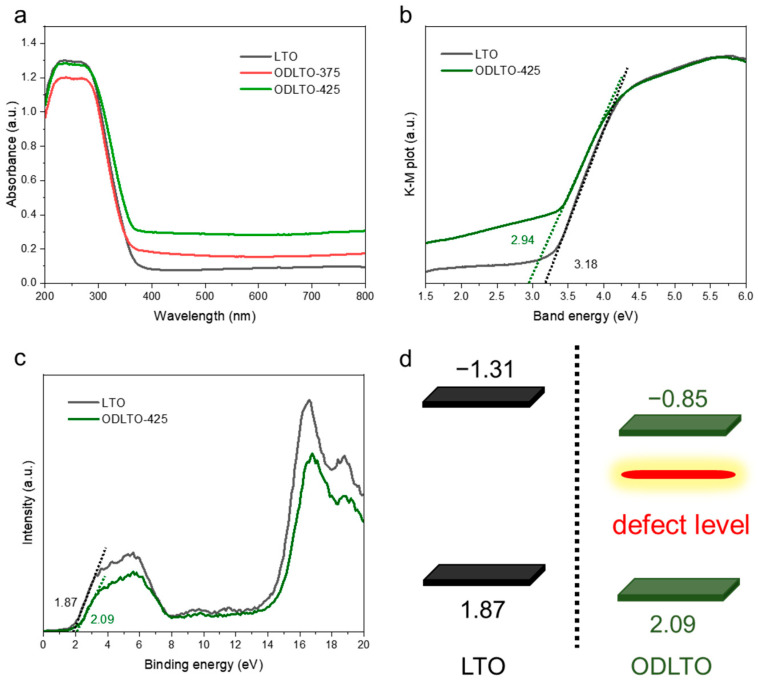
(**a**) UV-vis diffuse reflectance spectra. (**b**) Tauc plots. (**c**) VB-XPS spectra and (**d**) band structure of pristine LTO and ODLTO-425 samples.

**Figure 5 molecules-28-05792-f005:**
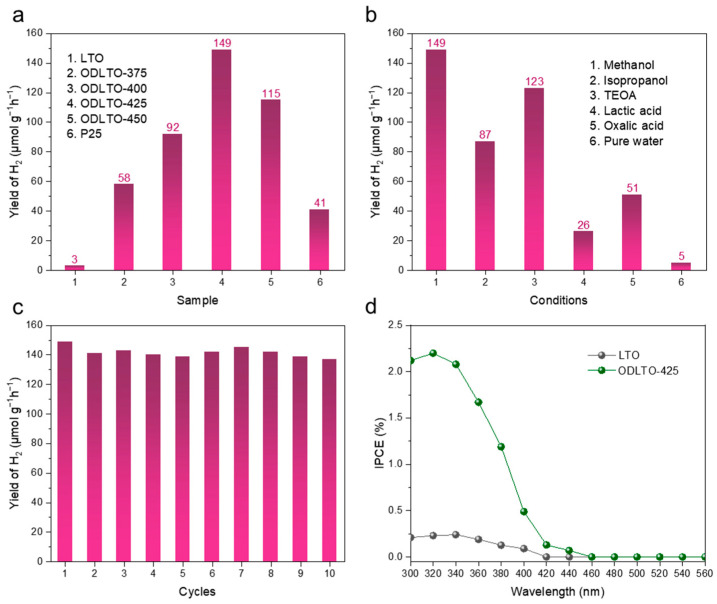
(**a**) H_2_ evolution rate of pristine LTO and serial ODLTO samples; (**b**) photocatalytic H_2_ evolution of ODLTO-425 sample under different sacrificial agents; (**c**) H_2_ recycling production experiments (10 runs) through with ODLTO-425 sample; (**d**) IPCE plots of the pristine LTO and serial ODLTO-425 sample under different wavelengths.

**Figure 6 molecules-28-05792-f006:**
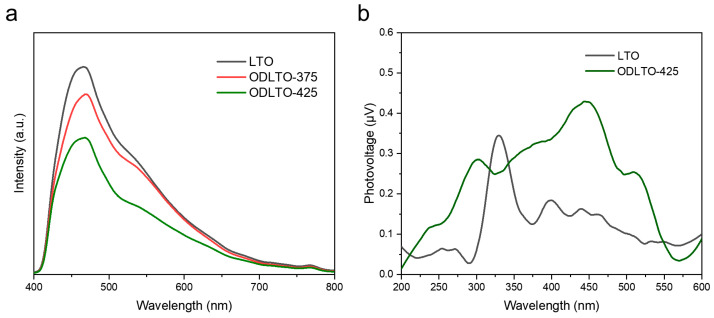
(**a**) PL and (**b**) SPV signals of the pristine LTO and ODLTO-425 samples.

## Data Availability

The data presented in this study are available in this article.

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
