# Peer review of "Oxygen Vacancies Defective La2Ti2O7 Nanosheets Enhanced Photocatalytic Activity of Hydrogen Evolution under Visible Light Irradiation"

_molecules, 2023, doi:10.3390/molecules28155792_

Round 1

Reviewer 1 Report

This work reports the introduction of oxygen vacancies on La2Ti2O7 nanosheets by a controlled solid-state reaction. Further, the La2Ti2O7 nanosheets with engineered oxygen vacancies have demonstrated an exceptional photocatalytic capacity for hydrogen production under visible light irradiation. This article is suitable for publication in Molecules after addressing the following comments:

1. The details of Photoelectrochemical Measurements (IPCE) should be provided.

2. The introduction lacks the key information on the background of vacancy engineering to highlight the benefits of the proposed photocatalyst more.

3. While LED light source has a tendency of inducing photothermal effect to the reaction system, the reaction temperature should also be determined in the section 3.3.

4. There are some typo and grammar issues in this manuscript, and the authors are suggested to check them carefully.

Author Response

Response to Reviewer #1

Dear Reviewer #1,

Thank you very much for your comments and suggestions on our manuscript. Accordingly, we have revised our manuscript. The detailed corrections are listed below point by point.

  1. The details of Photoelectrochemical Measurements (IPCE) should be provided.

Response: We thank the reviewer for the suggestion. Accordingly, we have made the following changes to the manuscript: “A conventional three-electrode process was used to investigate the photoelectrochemical properties of samples in a quartz cell. The FTO photoanode deposited STO samples, Hg/Hg2Cl2, and Pt foil electrode acted as the working electrode, reference electrode, and counter electrode, respectively. 1.0 M NaOH aqueous solution was used as the electrolyte. The photoanode was illuminated by a 300 W Xe lamp. The illuminated area was about 1 × 1 cm2.”.

  1. The introduction lacks the key information on the background of vacancy engineering to highlight the benefits of the proposed photocatalyst more.

Response: We thank the reviewer for raising the question. Accordingly, we have made the following changes to the manuscript: “Recently, it's been noted that these vacancies, also known as oxygen vacancy defect (OD), can augment solar light harvesting in semiconductors. This enhancement is achieved by narrowing the band gap and acting as active sites to boost carrier separation efficiency, thereby resulting in improved water splitting efficiency. Incorporating surface oxygen vacancies can generate new defect levels between the La2Ti2O7's valence band maximum (VBM) and conduction band minimum (CBM). These new levels are advantageous as they draw more towards visible-light absorption, thereby augmenting solar harvesting. Moreover, it's proposed that a high concentration of defects amassing at the contact interface of composites is advantageous for forming ohmic contact. This is because the surface oxygen defects can serve as adsorption sites and trapping centers for photoinduced charges. Therefore, management of oxygen defects on the surface of the photocatalyst could be instrumental for enhancing its photocatalytic properties.”.

  1. While LED light source has a tendency of inducing photothermal effect to the reaction system, the reaction temperature should also be determined in the section 3.3.

Response: Thank you very much for your suggestion. We have added the experimental details into the revised manuscript: “The H2 generation experiment was tested in a 435 mL sealed quartz three-necked Pyrex flask. The bottle was kept in a temperature-controlled air bath at 25 ± 0.5 °C with wind flow and was irradiated using a four 3 W LED lamp (420 nm).”

  1. There are some typo and grammar issues in this manuscript, and the authors are suggested to check them carefully.

Response: We thank the reviewer for the helpful suggestion. Accordingly, we have carefully checked the entire manuscript and corrected the typo and grammar errors.

Reviewer 2 Report

This work portrays an optimization of the catalyst for the formation of hydrogen. The work aims to increase the concentration of oxygen vacancies in the material to enhance the catalytic properties of the material. The authors clearly achieved this objective and demonstrated it to be an article with the potential to be published in this journal. In order to increase the quality of the article, I suggest the following modifications or additions to the article (I will also ask some questions to clarify some points):

1) Why do you use 20 mg of the samples?

2) All the manuscript needs to be checked in terms of number in subscript or superscript position.

3) Have the preparation conditions been optimized? Why were some temperatures fixed?

4) Why do hydrogen yields through cycles go up and down? Is it within the experimental error?

Author Response

Response to Reviewer #2

Dear Reviewer #2,

Thank you very much for your comments and suggestions on our manuscript. Accordingly, we have revised our manuscript. The detailed corrections are listed below point by point.

  1. Why do you use 20 mg of the samples?

Response: We thank the reviewer for the valuable suggestion. According to relevant literature reports, a typical 20 mg photocatalyst is used in photocatalytic hydrogen production experiments. Under these conditions, both sufficient dispersion of the catalyst and suitable calculation of yield and quantum efficiency can be ensured.

  1. All the manuscript needs to be checked in terms of number in subscript or superscript position.

Response: We thank the reviewer for the comment. We have checked and modified all the subscripts and superscripts in this manuscript.

  1. Have the preparation conditions been optimized? Why were some temperatures fixed?

Response: We thank the reviewer for the valuable comment. According to relevant literature, the formation of oxygen vacancies in perovskite photocatalytic systems usually requires maintaining a certain temperature during the annealing process. In this paper, we selected several specific temperatures for annealing, and the temperature variation led to the formation of different concentrations of oxygen vacancies.

  1. Why do hydrogen yields through cycles go up and down? Is it within the experimental error?

Response: We thank the reviewer for raising the question. I Typically, a gradual decline in performance during cyclic experiments is a common trend, but it is also quite common to observe higher performance in certain individual experiments. In conclusion, fluctuations in performance during cyclic experiments are considered a normal experimental phenomenon.

Reviewer 3 Report

1.    This study seems interesting. The experiments are well presented, and the results have value for practitioners.

2.    Figure 3,4, 5, and 6—Please more explanation

3.    Some leading works regarding “visible light irradiation” should be discussed in the introduction.

1.Chen, GC., Huang, WT., Lee, PC. et al. In situ precipitation 3D printing of highly ordered silver cluster–silver chloride photocatalysts. Int J Adv Manuf Technol 126, 797–811 (2023).

2.Yang, ZR., Lee, PC., Kuo, CY. et al. Application of high specific surface area Ag/AgCl/TiO2 coupled photocatalyst fabricated by fused filament fabrication. Int J Adv Manuf Technol 120, 4539–4550 (2022).

3.Liu, CH., Huang, CY., Lee, PC. et al. AgCl-based selective laser melting photocatalytic module for degradation of azo dye and E. coli. Int J Adv Manuf Technol 115, 1127–1138 (2021).

4.Obilor, A.F., Pacella, M., Wilson, A. et al. Micro-texturing of polymer surfaces using lasers: a review. Int J Adv Manuf Technol 120, 103–135 (2022).

5.Fan, X., Gao, X., Liu, G. et al. Research and prospect of welding monitoring technology based on machine vision. Int J Adv Manuf Technol 115, 3365–3391 (2021).

6.Gisario, A., Barletta, M. & Veniali, F. Laser polishing: a review of a constantly growing technology in the surface finishing of components made by additive manufacturing. Int J Adv Manuf Technol 120, 1433–1472 (2022).

7.Liu, K., Zhang, Z., Sun, H. et al. Fabrication and mechanical properties of triply period minimal surface porous alumina ceramics based on Digital Light Processing 3D printing technology. Int J Adv Manuf Technol (2023). https://doi.org/10.1007/s00170-023-11164-z

8.Lukong, V.T., Ukoba, K. & Jen, TC. Review of self-cleaning TiO2 thin films deposited with spin coating. Int J Adv Manuf Technol 122, 3525–3546 (2022).

9.Belekbir, S.; El Azzouzi, M.; Rodríguez-Lorenzo, L.; El Hamidi, A.; Santaballa, J.A.; Canle, M. Cobalt Impregnation on Titania Photocatalysts Enhances Vis Phenol Photodegradation. Materials 2023, 16, 4134.

10.Kedves, E.-Z.; Bárdos, E.; Ravasz, A.; Tóth, Z.-R.; Mihálydeákpál, S.; Kovács, Z.; Pap, Z.; Baia, L. Photoinhibitive Properties of α-MoO3 on Its Composites with TiO2, ZnO, BiOI, AgBr, and Cu2O. Materials 2023, 16, 3621.

11.Grochowska, K.; Haryński, Ł.; Karczewski, J.; Jurak, K.; Siuzdak, K. Scanning with Laser Beam over the TiO2 Nanotubes Covered with Thin Chromium Layers towards the Activation of the Material under the Visible Light. Materials 2023, 16, 2572.

12.Alharbi, R.; Alharbi, E.; Al-Haj Ali, S.N.; Farah, R.I. Thickness-Dependent Light Transmittance and Temperature Rise in Dual-Cure Bioactive and Light-Cure Bulk-Fill Composite Resins. Polymers 2023, 15, 2837.

13.Coromelci, C.G.; Turcu, E.; Doroftei, F.; Palamaru, M.N.; Ignat, M. Conjugated Polymer Modifying TiO2 Performance for Visible-Light Photodegradation of Organics. Polymers 2023, 15, 2805.

14.Coderch, G.; Cordoba, A.; Ramírez, O.; Bonardd, S.; Leiva, A.; Häring, M.; Díaz Díaz, D.; Saldias, C. Effects of the Solvent Vapor Exposure on the Optical Properties and Photocatalytic Behavior of Cellulose Acetate/Perylene Free-Standing Films. Polymers 2023, 15, 2787.

15.Jiang, S.; Yin, M.; Ren, H.; Qin, Y.; Wang, W.; Wang, Q.; Li, X. Novel CuMgAlTi-LDH Photocatalyst for Efficient Degradation of Microplastics under Visible Light Irradiation. Polymers 2023, 15, 2347.

4.    Finally, I would suggest the author to address the questions above in the revision. I am pleased to review the revised manuscript.

1.    This study seems interesting. The experiments are well presented, and the results have value for practitioners.

2.    Figure 3,4, 5, and 6—Please more explanation

3.    Some leading works regarding “visible light irradiation” should be discussed in the introduction.

1.Chen, GC., Huang, WT., Lee, PC. et al. In situ precipitation 3D printing of highly ordered silver cluster–silver chloride photocatalysts. Int J Adv Manuf Technol 126, 797–811 (2023).

2.Yang, ZR., Lee, PC., Kuo, CY. et al. Application of high specific surface area Ag/AgCl/TiO2 coupled photocatalyst fabricated by fused filament fabrication. Int J Adv Manuf Technol 120, 4539–4550 (2022).

3.Liu, CH., Huang, CY., Lee, PC. et al. AgCl-based selective laser melting photocatalytic module for degradation of azo dye and E. coli. Int J Adv Manuf Technol 115, 1127–1138 (2021).

4.Obilor, A.F., Pacella, M., Wilson, A. et al. Micro-texturing of polymer surfaces using lasers: a review. Int J Adv Manuf Technol 120, 103–135 (2022).

5.Fan, X., Gao, X., Liu, G. et al. Research and prospect of welding monitoring technology based on machine vision. Int J Adv Manuf Technol 115, 3365–3391 (2021).

6.Gisario, A., Barletta, M. & Veniali, F. Laser polishing: a review of a constantly growing technology in the surface finishing of components made by additive manufacturing. Int J Adv Manuf Technol 120, 1433–1472 (2022).

7.Liu, K., Zhang, Z., Sun, H. et al. Fabrication and mechanical properties of triply period minimal surface porous alumina ceramics based on Digital Light Processing 3D printing technology. Int J Adv Manuf Technol (2023). https://doi.org/10.1007/s00170-023-11164-z

8.Lukong, V.T., Ukoba, K. & Jen, TC. Review of self-cleaning TiO2 thin films deposited with spin coating. Int J Adv Manuf Technol 122, 3525–3546 (2022).

9.Belekbir, S.; El Azzouzi, M.; Rodríguez-Lorenzo, L.; El Hamidi, A.; Santaballa, J.A.; Canle, M. Cobalt Impregnation on Titania Photocatalysts Enhances Vis Phenol Photodegradation. Materials 2023, 16, 4134.

10.Kedves, E.-Z.; Bárdos, E.; Ravasz, A.; Tóth, Z.-R.; Mihálydeákpál, S.; Kovács, Z.; Pap, Z.; Baia, L. Photoinhibitive Properties of α-MoO3 on Its Composites with TiO2, ZnO, BiOI, AgBr, and Cu2O. Materials 2023, 16, 3621.

11.Grochowska, K.; Haryński, Ł.; Karczewski, J.; Jurak, K.; Siuzdak, K. Scanning with Laser Beam over the TiO2 Nanotubes Covered with Thin Chromium Layers towards the Activation of the Material under the Visible Light. Materials 2023, 16, 2572.

12.Alharbi, R.; Alharbi, E.; Al-Haj Ali, S.N.; Farah, R.I. Thickness-Dependent Light Transmittance and Temperature Rise in Dual-Cure Bioactive and Light-Cure Bulk-Fill Composite Resins. Polymers 2023, 15, 2837.

13.Coromelci, C.G.; Turcu, E.; Doroftei, F.; Palamaru, M.N.; Ignat, M. Conjugated Polymer Modifying TiO2 Performance for Visible-Light Photodegradation of Organics. Polymers 2023, 15, 2805.

14.Coderch, G.; Cordoba, A.; Ramírez, O.; Bonardd, S.; Leiva, A.; Häring, M.; Díaz Díaz, D.; Saldias, C. Effects of the Solvent Vapor Exposure on the Optical Properties and Photocatalytic Behavior of Cellulose Acetate/Perylene Free-Standing Films. Polymers 2023, 15, 2787.

15.Jiang, S.; Yin, M.; Ren, H.; Qin, Y.; Wang, W.; Wang, Q.; Li, X. Novel CuMgAlTi-LDH Photocatalyst for Efficient Degradation of Microplastics under Visible Light Irradiation. Polymers 2023, 15, 2347.

4.    Finally, I would suggest the author to address the questions above in the revision. I am pleased to review the revised manuscript.

Author Response

Response to Reviewer #3

Dear Reviewer #3,

Thank you very much for your comments and suggestions on our manuscript. Accordingly, we have revised our manuscript. The detailed corrections are listed below point by point.

  1. This study seems interesting. The experiments are well presented, and the results have value for practitioners.

Response: We thank the reviewer for the suggestion.

  1. Figure 3,4, 5, and 6—Please more explanation

Response: We thank the reviewer for raising the question. The additional explanations related to Figures 3, 4, 5, and 6 have been added in the revised manuscript.

  1. Some leading works regarding “visible light irradiation” should be discussed in the introduction.

Response: Thank you very much for your suggestion. I have already added the Refs about visible light irradiation in above-mentioned references to the revised manuscript.

  1. Finally, I would suggest the author to address the questions above in the revision. I am pleased to review the revised manuscript.

Response: We thank the reviewer for the helpful suggestion. Accordingly, we have carefully checked the entire manuscript and corrected the typo and grammar errors. I have already added the above-mentioned references to the revised manuscript.

Round 2

Reviewer 3 Report

The revised manuscript now can be accepted in the journal for publication as the authors have incorporated all the suggestions.

The revised manuscript now can be accepted in the journal for publication as the authors have incorporated all the suggestions.